# Changes and Characteristics of Green Infrastructure Network Based on Spatio-Temporal Priority

Xifan Chen [1], Lihua Xu [1,*], Rusong Zhu [2], Qiwei Ma [1], Yijun Shi [1] and Zhangwei Lu [1]

1    School of Landscape Architecture and Architecture, Zhejiang Agriculture and Forestry University, Hangzhou 310000, China; chenxifan@stu.zafu.edu.cn (X.C.); maqiwei@zafu.edu.cn (Q.M.); yijun_shi@zafu.edu.cn (Y.S.); zhwlu@zafu.edu.cn (Z.L.)
2    Zhejiang Yuanzhuo Science and Technology Company Limited, Hangzhou 310000, China; zhurs1981@126.com
*    Correspondence: xulihua@zafu.edu.cn; Tel.: +86-13805799690

**Abstract:** With advancements in urbanization, natural lands are constantly being encroached upon by artificial impervious surfaces, leading to serious ecosystem damage. Calls for Green Infrastructure to address urban environmental issues and resource reallocation are growing. How to optimize Green Infrastructure networks are becoming increasingly important under rapid urbanization. In this study, we used the main city zone in Hangzhou as the study area, and we extracted 2000, 2010 and 2020 land-use data. We used morphological spatial pattern analysis to identify Green Infrastructure landscape types and further extract Green Infrastructure elements. We identified the spatial priority of Green Infrastructure network elements through landscape connectivity evaluation according to ecological importance and development vulnerability. After the construction of a Green Infrastructure network, we analyzed its spatio-temporal characteristics to determine the Green Infrastructure network's spatial priority. Through spatial prioritization, the gradual construction and optimization of Green Infrastructure networks will help to improve urban green spaces in stages. Smartly coordinating urban growth and ecological protection based on Green Infrastructure spatial prioritization may help improve urban living environments and enhance sustainable urban development capabilities. In conclusion, sources dominate corridors and codes are changing. If sources are fragmented, the integration degree decreases and the first-level source advantage is weakened. The corridor morphology continuously develops, and the corridor structure stabilizes. Second-level corridors gradually replace third-level corridors to guide Green Infrastructure network structure development. Codes present a scatter distribution and tend to average, closely following corridor change.

**Keywords:** Green Infrastructure; Morphological Spatial Pattern Analysis; spatial priority; spatio-temporal characteristics

## 1. Introduction

Nowadays, rapid urbanization is resulting in numerous urban land-use, ecological, and environmental issues that are seriously threating ecological security in urban areas [1,2]. Sustained biodiversity loss, habitat fragmentation, environmental degradation, air pollution [3] and climate change [4–6] are leading to declining ecosystem services [7] and challenging sustainable urban development. We should urgently reduce negative anthropogenic pressure on the natural environment and promote ecological restoration. Many cities worldwide are looking to use re-greening strategies to help reverse urbanization patterns that aggravate environmental issues [8–10]. Therefore, international attention to urban green space is continuously intensifying, and calls for Green Infrastructure to address urban environmental issues and resource reallocation are growing. Research has confirmed that Green Infrastructures (GIs) are the main carriers of ecosystem services and have an

important impact on human well-being [1,11,12]. Green infrastructure is generated with green ecological space development [13]. Frederick Law Olmsted [14,15] first proposed Green Infrastructure in the 1880s in natural design vision, an idea that received widespread attention. Green infrastructure is a strategically planned network of natural and semi-natural areas [15,16], helping to maintain environment quality and providing residents with high-quality ecosystem services [11,17,18]. In 1999, the United States adopted Green Infrastructure as a tool to help achieve sustainable development in the future [15]. In recent years, international scholars, scientists and politicians have paid great attention to Green Infrastructure applications in urban services, such as stormwater management, climate regulation, and urban greening [19,20]. The concept of Green Infrastructure describes the interdependence of land conservation and land development. Green infrastructure has gradually become an integral part of spatial planning and territorial development. Green infrastructure helps protect non-built-up land by highlighting societal benefits related to green space [21]. However, to achieve these multifunctional goals, both the quantity and quality of urban green space and urban marginal green space must be considered in the planning process [18], in which Green Infrastructure development is fundamental.

Though Green Infrastructure planning has gradually become an important national strategy for providing ecosystem services and coordination in promoting green economic development for cities [1,21–23], there are still several unresolved problems limiting a more general implementation. In previous Green Infrastructure studies and related literature, experts generally focused on Green Infrastructure network construction [12,24,25] (Table 1). The most straightforward way to outline Green Infrastructure is to select the specific land-use and land-cover (LULC) types that comprise a network [26,27]. Other authors adopted a spatial overlay approach [28], which formed a mainstream research framework of "source determination–resistance surface setting–corridor construction–code identification–GI network construction". Scholars consider that the importance of each GI element is different, so a series of studies with different emphases have been conducted. Some experts have focused on source determination and corridor construction to try to realize the smooth flow of ecological process [24,29,30]. Other experts think that GI key code identification plays an important role, and it should be applied to urban ecological network planning as soon as possible [31,32]. In the previous research, Green Infrastructure element extraction methods mainly included morphological spatial pattern analysis (MSPA), the InVEST model, the minimum cumulative resistance (MCR) model, the circuit theory model [33], and Linkage Mapper [12,28,34]. MSPA focuses on measuring structural connectivity and enables the quick and efficient identification of ecological elements [35]. By dividing ecological land into seven landscape types, the type and structure of ecological land can be more accurately identified, which is conducive to GI network construction.

**Table 1.** The content and methods of Green Infrastructure research.

| Research Aspect | Research Content/Methods |
| --- | --- |
| Green Infrastructure application in urban services | Stormwater management, climate regulation, urban greening, and so on |
| Green Infrastructure network construction | Select the specific LULC types, spatial overlay approach |
| Green Infrastructure elements extraction | MSPA, InVEST model, MCR model, circuit theory model, Linkage Mapper, and so on |
| Green Infrastructure network evaluation | Ecological connectivity index, landscape pattern index, landscape connectivity index, and so on |
| Green Infrastructure network prioritization | GIS-based multicriteria evaluation methods, the progressive Green Infrastructure zoning method, the participatory mapping method, the SCP method, and so on |

Research has revealed that the focus of GI implementation in European countries is always to strengthen ecological network construction [36]. The number of degraded ecosystems in cities is numerous, and ecological resources and construction funds are

limited. Indiscriminately constructing Green Infrastructure is unrealistic and difficult to implement. At present, Green Infrastructure networks are mainly comprehensively evaluated with an ecological connectivity index, a landscape pattern index [37], a landscape connectivity index, and other methods. It is expected that the optimization path of Green Infrastructure network construction can be based on overall change of plot indexes as well. However, different management strategies for different Green Infrastructure spatial priorities can ensure that limited land resources and public funds are directed to where they are truly needed. Green Infrastructure spatial prioritization is based on the spatial hierarchy of ecological importance and development vulnerability, and it intended to be used to more accurately prioritize Green Infrastructure elements [38]. It is an effective measure to coordinate smart city growth and protection. Spatial prioritization can be used to not only scientifically propose Green Infrastructure network planning directions but also propose constructive suggestions for protection, restoration, and development. At present, the methods of prioritizing Green Infrastructure networks are mainly based on geographic information system (GIS)-based multicriteria evaluation methods, the progressive Green Infrastructure zoning method, the participatory mapping method, and the spatial conservation prioritization (SCP) method [39,40]. The authors of some studies developed software tools for spatial conservation prioritization aimed at biodiversity [41]. Ou X [39] coupled the comprehensive evaluation of multi-hazard risk and the SCP method to improve the adaptability of Green Infrastructure networks to multiple climate changes. In most previous research, the authors prioritized Green Infrastructure in planning units, not from the elements of GI network. Some studies were focused on prioritizing individual constituent elements and did not consider their overall impact on Green Infrastructure networks.

Landscape connectivity is broadly defined as the degree to which the landscape facilitates or hinders movement among resource patches [42]. Several authors have described landscape connectivity as the core principle of Green Infrastructure [43,44], which is one of the most frequently mentioned principles in the literature [23]. It is an important indicator of landscape pattern and function [42], affecting species richness and migration processes. The movement of organisms, genetic interchange, and other ecological flows are essential for species survival and biodiversity conservation in general [45]. Ahern [46] described landscape connectivity as the key principle for green space and greenway development in Green Infrastructure research in 1995. Spatial planning and land management are essential for urban ecology protection from the landscape connectivity perspective [47]. More and more research is now focused on how the landscape connectivity principle is affirmatively implemented and evaluated [44,48]. Evaluating landscape connectivity by using effective models and measurement methods is necessary to appropriately maintain and improve ecological networks [24]. Green Infrastructure spatial prioritization according to landscape connectivity can be used to realize the Green Infrastructure ecological construction process in an orderly and efficient manner.

Spatio-temporal analysis helps policy-makers guarantee sustainable development and understand the dynamics of the changing environment [49,50]. Kemarau R A and Chen C [51,52] investigated the spatio-temporal pattern changes of urban green space. Most urban green space change studies are characterized by analyses of green space change characteristics and patterns in particular cities or regions [53,54]. Dynamic research on urban green spaces enables the continuous and successful management of the urban environment, specifically regarding green space change in urban planning and decision making [55,56]. Detailed studies in which the authors analyze the spatio-temporal characteristics of urban Green Infrastructure are still relatively lacking. A literature review confirmed that most researchers missed the temporal aspect of Green Infrastructure and focused on the Green Infrastructure concept, development, and planning—topics that are concentrated around Green Infrastructure's value and application in practice. The study of Green Infrastructure's spatio-temporal variation characteristics can be used to effectively cope with ecological problems caused by rapid urbanization [57,58] and to optimize urban Green Infrastructure networks. It can also be used to improve the knowledge of planners and help policymakers

understand the green spaces recognized in urban areas for strategic planning, which have great significance in urban green space system planning [20].

In this study, we mainly focused on urbanization areas and optimized a Green Infrastructure network through the identification of spatial priority. In China, cities with strong rapid urbanizationinclude Beijing, Shanghai, and Hangzhou. Here, combined with Hangzhou urban construction and development, we used a typical Hangzhou urbanization area as the study object. The innovation of the research lies in the construction of a Green Infrastructure network carrying spatial priority information through the spatial prioritization of GI elements. We focused on the spatio-temporal characteristics and spatial priority of the Green Infrastructure network in order to seek solutions for smart and coordinated urban growth and ecological protection in developing country urbanized areas. The overall contents are as follows:

1. We used MSPA to identify landscape types and combined other methods to extract Green Infrastructure elements at different time periods over the past 20 years.

2. We divided the spatial priority of different Green Infrastructure elements based on various methods of landscape connectivity, and then we constructed a GI network.

3. We quantitatively analyzed the spatio-temporal characteristics of Green Infrastructure network features to provide effective construction timing suggestions for scientific Green Infrastructure network construction.

## 2. Materials and Methods

### 2.1. Study Area

We studied a city zone (coordinates: $118°21'$ E, $29°11'$N–$120°30'$E, $30°33'$ N) in Hangzhou, with a 3355.53 km$^2$ area, that accounts for 3.27% of Zhejiang Province (Figure 1). Hangzhou is located in the north of Zhejiang Province, with flat terrain and a dense river network. Hangzhou's Green Infrastructure mainly surrounds the outside of the city, in addition to some scattered sections in the center. The Qiantang River and the Grand Beijing-to-Hangzhou Canal pass through Hangzhou, and Hangzhou's green space is rich. The core of Hangzhou is the West Lake Scenic Area, and the axis belt is the Grand Beijing-to-Hangzhou Canal. Urban and suburban green spaces combine to form the Green Infrastructure network, creating an ecological circle layer inside and outside the city. Hangzhou has rapidly developed since the 20th century due to population increases and continuous urbanization acceleration. According to data, in 2020, Hangzhou's population urbanization rate rose to 83.29%, the total GDP increased by 3.9% over the previous year, and the built-up area was 567.32 km$^2$. Against the background of limited land resources, ecological risks caused by rapid urbanization are particularly prominent in the main city zone. Therefore, we used Hangzhou as the research object. This study will have significance for the spatio-temporal characteristic analyses of Green Infrastructure networks in other regions, and the results can serve as an important reference and basis for Green Infrastructure network construction.

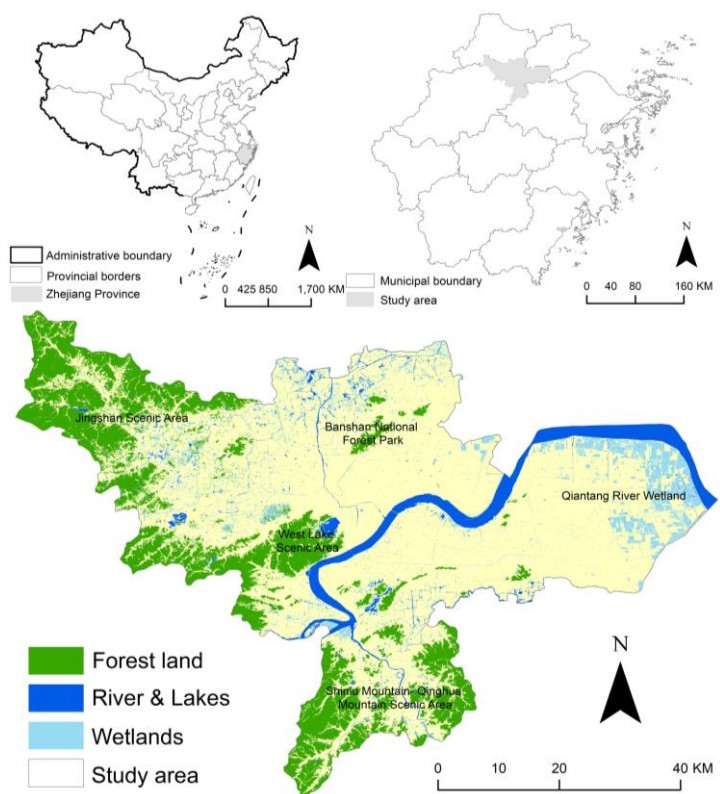

**Figure 1.** The study area location.

### 2.2. Data Sources and Processing

The data collected in this study contained image and text materials including geographic information and remote sensing image data. We downloaded remote sensing image data from Landsat TM/OLI in the study area on 17 September 2000; 31 October 2010; and 8 September 2020 (source: US Geological Survey). To avoid seasonal changes effects in plants, remote sensing images were collected in September. If the cloud cover in September exceeded 5%, images of their adjacent months were used. The ENVI 5.3 software was used to preprocess image data through geometric correction, registration, radiometric correction, stitching, and clipping, among other techniques. According to land-use data, the study area is divided into six categories: forest land, farmland, buildable land, river and lakes, wetlands, and unused land (Figure 2). The kappa coefficients (K) of three years of land-use classification results were greater than 0.8. From 2000 to 2020, Hangzhou was in a rapid development period, and the urban built-up area was rapidly expanding. The excessive encroachment of buildable land on farmland led to drastic changes in land use. Forest land, river and lakes, and wetlands were translated into GI elements, and buildable land, unused land, and farmland were translated into non-GI elements.

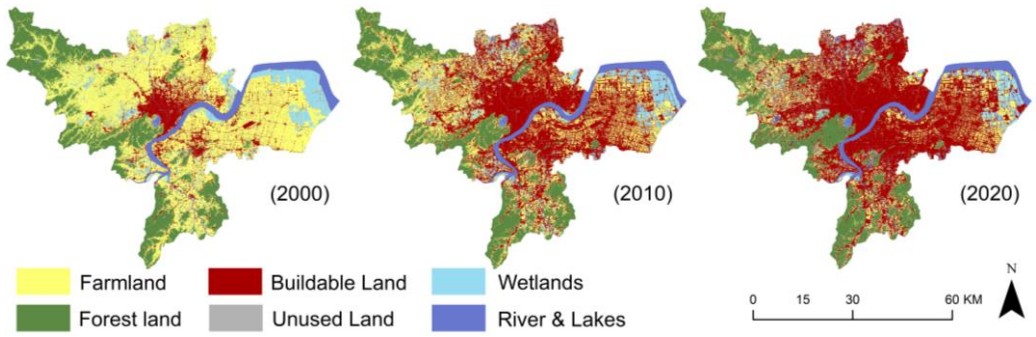

**Figure 2.** Land use of the study area in 2000, 2010, and 2020.

*2.3. Methods*

GI refers to a city's important ecological composition and is composed of sources, corridors, and codes. In this study, we spatially prioritized GI elements and studied the spatio-temporal characteristics of GI network features. A technical flowchart of this study is provided in Figure 3 and consists of four technical steps. First, on the basis of pretreatment, we extracted Green Infrastructure landscape types using MSPA. We further extracted sources, corridors, and nodes. Second, we divided the spatial prioritization of GI elements with different characteristics according to various methods of landscape connectivity evaluation. Third, we built a Green Infrastructure network carrying spatial priority information. Finally, we analyzed Green Infrastructure network spatial priority changes, which can be used to provide suggestions for the timing of Green Infrastructure network construction.

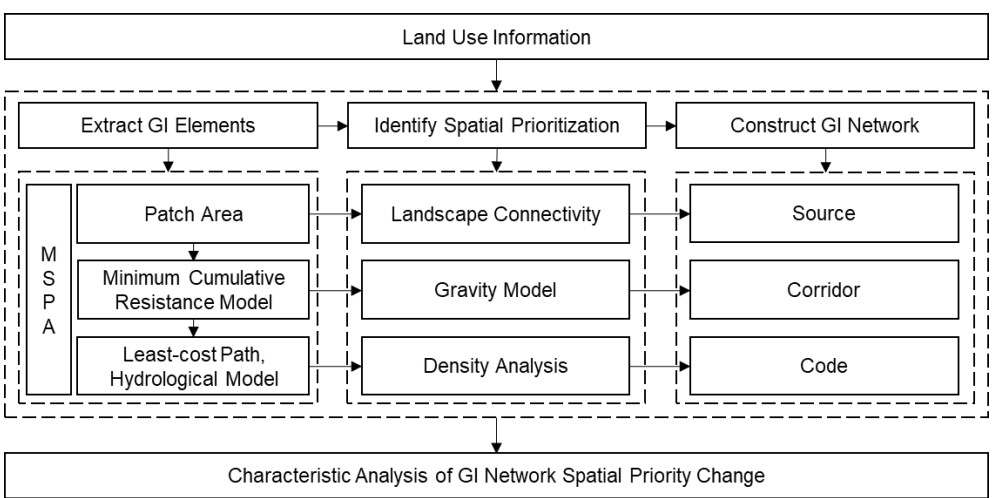

**Figure 3.** The technical flowchart of this study.

2.3.1. Extract GI Elements

Based on mathematical morphology principle, MSPA uses corrosion, expansion, opening, and closing operations to classify raster image cells from spatial morphology and structural connectivity perspectives [59]. The type of landscape acquired based on the MSPA has a clear scale effect and edge effect. Based on the foreground and background division of Green Infrastructure elements, comprehensive previous research [60,61], and multiple sets of parameters for experimental comparison, we used the eight-neighborhood rule and set the edge width to one, which allowed us to obtain detailed landscape information of the study area. We divided the foreground into seven landscape types with different functions and ecological meanings that did not contain each other [62,63]: branch, bridge, core, edge, islet, loop, and perforation (Figure 4). Core, branch, and bridge could be further extracted as sources and corridors.

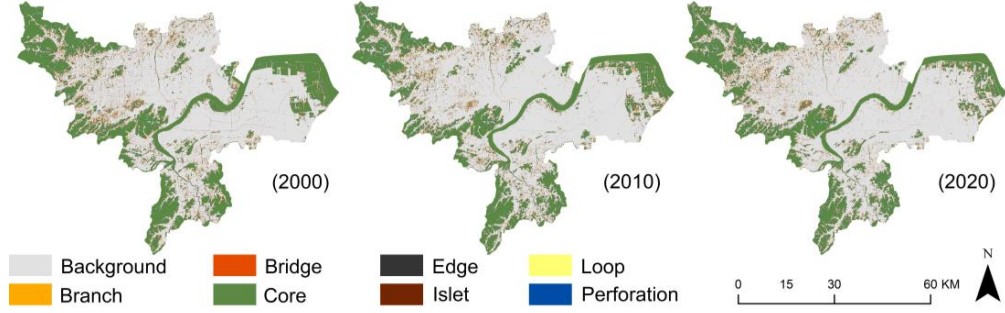

**Figure 4.** MSPA landscape types of the study area in 2000, 2010, and 2020.

According to patch functional characteristics, the larger the area, the better the connectivity, and the more suitable an area is for biological habitats, the higher its spatial priority. Considering patch area and connectivity in tandem with the distribution status and law, we screened cores with 10 ha or more of area as sources.

Surface resistance refers to an organism's willingness to move through a landscape [33]. MCR is used to calculate the total resistance that species need to overcome during the journey from "source" point to destination, generate a least-cost path (LCP), and realize potential corridor construction between multiple sources. Referring to previous research [64], we selected five factors covering natural environment and human activity interference: MSPA landscape types, land-use types, elevation, slope, and relief degree of land surface. We graded each factor, and the value standards and weights of each grade were determined with the analytic hierarchy process (Table 2).

$$MCR = fmin \sum_{j=n}^{i=m} \left( D_{ij} \times R_i \right) \qquad (1)$$

where *MCR* represents the minimum cumulative resistance value, $f$ represents the pending monotonic increment function, $D_{ij}$ represents the spatial distance from source point $j$ to space unit I, $R_i$ represents the space unit $i$'s resistance coefficient, min represents the cumulative minimum value of resistance surface generated by different levels of "sources", $m$ is the resistance surface number, and $n$ is the source number [35].

**Table 2.** Resistance values and coefficients of each factor.

| Resistance Type | Resistance Coefficient | Resistance Factor | Resistance Value |
|---|---|---|---|
| MSPA Landscape Type | 0.48 | Core | 5 |
| | | Bridge | 10 |
| | | Loop | 20 |
| | | Branch | 30 |
| | | Islet | 50 |
| | | Edge | 60 |
| | | Perforation | 70 |
| | | Background | 100 |
| Land-Use Type | 0.27 | Forest land | 1 |
| | | Farmland | 30 |
| | | Unused Land | 50 |
| | | Wetlands | 60 |
| | | River and Lakes | 70 |
| | | Buildable Land | 100 |
| Elevation | 0.12 | h1 < 200 m | 1 |
| | | $200 \leq$ h1 < 400 m | 20 |
| | | $400 \leq$ h1 < 800 m | 60 |
| | | $800 \leq$ h1 < 1000 m | 80 |
| | | h1 $\geq$ 1000 m | 100 |
| Slope | 0.08 | i < 8° | 1 |
| | | $8° \leq$ 1 < 15° | 20 |
| | | $15° \leq$ 1 < 25° | 60 |
| | | $25° \leq$ 1 < 35° | 80 |
| | | i $\geq$ 35° | 100 |
| Relief Degree of Land Surface | 0.05 | h2 < 15 | 1 |
| | | $15 \leq$ h2 < 30 | 20 |
| | | $30 \leq$ h2 < 60 | 60 |
| | | $60 \leq$ h2 < 90 | 80 |
| | | h2 $\geq$ 90 | 100 |

Codes are important "ecological stepping stones" for interconnection between sources, promoting the realization of Green Infrastructure networks from structural to functional connectivity. Codes are the transit stations for material exchange, generally in connecting

corridors vulnerable areas. We combined two methods to extract Green Infrastructure codes in this study. One was to extract intersection points between LCPs, which could provide multiple paths for species migration. The other was to extract intersections between LCPs and maximum-cost paths, which are the most vulnerable location of ecological corridors, with the help of a hydrological model.

### 2.3.2. Identify Spatial Prioritization

The landscape connectivity of sources was evaluated by calculating landscape indexes, such as probability connectivity (*PC*) and dependence of probability connectivity (*dPC*) [65].

$$PC = \frac{\sum_{i=1}^{n} \sum_{j=1}^{n} P_{ij}{}^{*} \cdot a_i \cdot a_j}{A_L{}^2} \tag{2}$$

$$dPC(\%) = 100 \cdot \frac{I - I_{remove}}{I} \tag{3}$$

where $n$ represents the total number of patches in the landscape; $a_i$ and $a_j$ are the patch $i$ and patch $j$ areas, respectively; $P_{ij}{}^{*}$ represents the maximum potential for species to spread directly in patch $i$ and patch $j$; $A_L$ represents the total landscape area; $I$ is an index value when all initially existing patches are present in the landscape; and $I_{remove}$ represents the index value after the removal of a single patch from the landscape [31].

Corridor importance is a comprehensive evaluation of ecological function and connecting source contributions. We used the gravity model to construct the interaction matrix between each source, and we quantitatively evaluated connecting corridors' relative importance. The greater the interaction force of the material exchange function that a corridor carried between sources, the higher the flow of carried ecological flow and the higher corridor spatial priority.

$$G_{ab} = \frac{N_a N_b}{D_{ab}{}^2} = \frac{\left(\frac{1}{P_a} \times \ln S_a\right)\left(\frac{1}{P_b} \times \ln S_b\right)}{\left(L_{ab}/L_{max}\right)^2} = \frac{L_{max}{}^2 \ln S_a \ln S_b}{L_{ab}{}^2 P_a P_b} \tag{4}$$

where $G_{ab}$ represents the interaction force between source $a$ and source $b$; $N_a$ and $N_b$ are weights; $D_{ab}$ represents *a* standardized value of potential corridor resistance between source $a$ and source $b$; $P_a$ and $P_b$ are the source $a$ and source $b$ resistance values, respectively; $S_a$ and $S_b$ are the source $a$ and source $b$ areas, respectively; $L_{ab}$ represents the cumulative resistance value of corridors between source $a$ and source $b$; and $L_{max}$ represents the maximum value of all corridors' cumulative resistance.

Codes were spatially prioritized through density analysis, which showed that the higher the density, the more important it was to improve Green Infrastructure network connectivity and the higher the spatial priority.

$$Density = \frac{1}{(radius)^2} \sum_{i=1}^{n} \left[ \frac{3}{\pi} \cdot pop_i \left( 1 - \left( \frac{dist_i}{radius} \right)^2 \right)^2 \right] \tag{5}$$

where $i = 1, ..., n$ represents the input point; $pop_i$ represents the point $i$ population field value; $dist_i$ represents the distance between point $i$ and (x, y) positions; and *radius* represents a given value.

## 3. Results and Analysis

### 3.1. GI Source Spatial Prioritization

According to the landscape connectivity results, *dPC* < 0.05 patches were used as a third-level source, $0.05 \leq dPC < 0.2$ were used as a second-level source, and $dPC \geq 0.2$ were used as a first-level source (Figure 5, Tables 3 and 4). No obvious changes in Hangzhou source spatial priority levels were observed. The edge of the city was found to change

significantly, and dominant sources were mainly distributed in the suburbs and the middle of the city. First- and second-level source advantages were shown to be higher than third-level sources. The number of first-level sources gradually increased from 57 to 83, and the area decreased from 74735.65 to 60065.74 ha. The number of second-level sources gradually increased from 39 to 68, and the area increased from 8229.01 to 105060.50 ha. The number of third-level sources gradually decreased from 187 to 161, and the area decreased from 4523.68 to 3941.51 ha. Some sources shrunk in area, but the dPC values were always much higher than other areas (that were still shown to be good habitat sources) such as the West Lake Scenic Area (B). Some source core patches maintained a high spatial priority, but the spatial priority of fragmented patches, such as the Qiantang River Wetland (C) and the Shiniu Mountain–Qinghua Mountain Scenic Area (D), decreased. Some second-level sources' spatial priority improved, and these sources were connected to first-level sources, such as the Jingshan Scenic Area (E). Some third-level source patches gradually dispersed and disappeared, such as the Northern and Central Isolated Areas such as the Banshan National Forest Park (A).

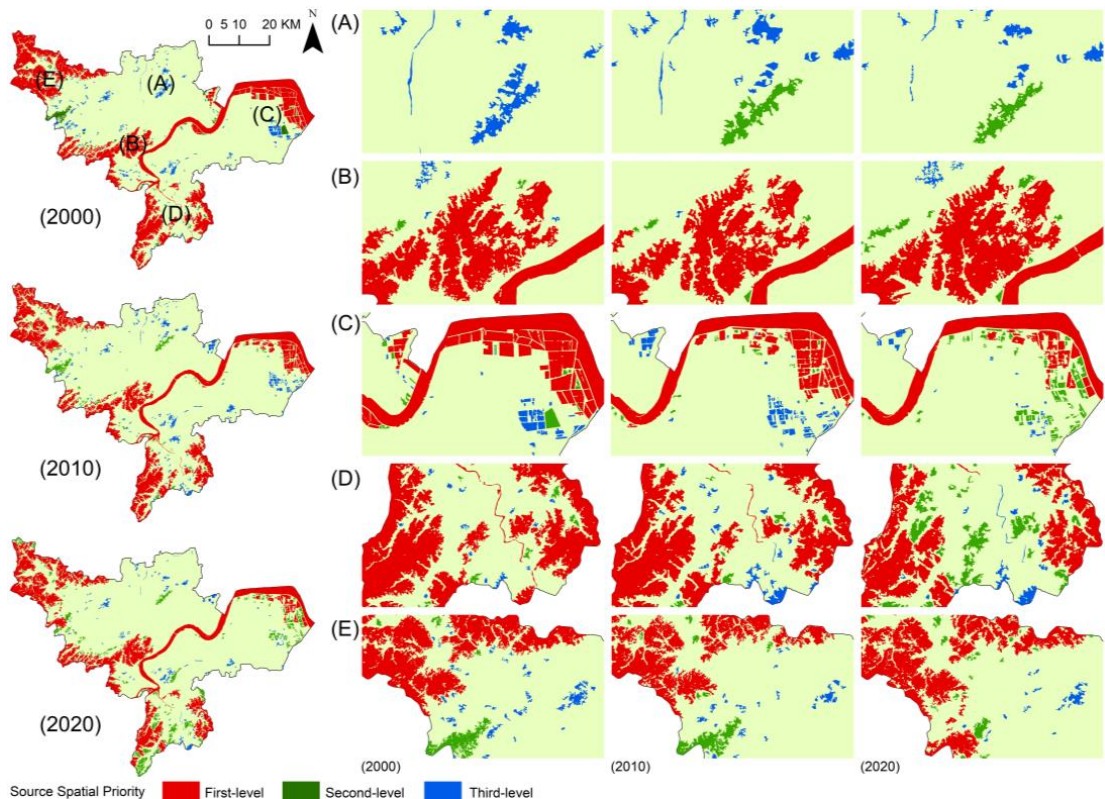

**Figure 5.** GI sources' spatial priority distribution at all levels in 2000, 2010, and 2020: (**A**) Banshan National Forest Park, (**B**) West Lake Scenic Area, (**C**) Qiantang River Wetland, (**D**) Shiniu Mountain–Qinghua Mountain Scenic Area, and (**E**) Jingshan Scenic Area.

**Table 3.** Division and definition of the spatial prioritization of different GI elements at each level.

|  | First-Level Spatial Prioritization | Second-Level Spatial Prioritization | Third-Level Spatial Prioritization |
|---|---|---|---|
| **Source** | dPC ≥ 0.2 | 0.05 ≤ dPC < 0.2 | dPC < 0.05 |
| **Corridor** | Gab ≥ 100 | 10 ≤ Gab < 100 | Gab < 10 |
| **Code** | Di ≥ 0.5 | 0.2 ≤ Di < 0.5 | Di < 0.2 |
| **Definition** | High Spatial or Temporal Prioritization Indices in Landscape Connectivity | Average Spatial or Temporal Prioritization Indices in Landscape Connectivity | Low Spatial or Temporal Prioritization Indices in Landscape Connectivity |

**Table 4.** The number of GI elements at all levels that changed in 2000, 2010, and 2020.

| | Source (Piece/ha) | | | Corridor (Strip/m) | | | Code (Piece) | | |
|---|---|---|---|---|---|---|---|---|---|
| | First-Level | Second-Level | Third-Level | First-Level | Second-Level | Third-Level | First-Level | Second-Level | Third-Level |
| **2000** | 57/74735.65 | 39/8229.01 | 187/4523.68 | 29/17270.50 | 31/88262.73 | 115/330152.76 | 22 | 39 | 6 |
| **2010** | 73/62163.95 | 55/7563.38 | 175/5549.41 | 21/19235.42 | 56/184329.71 | 103/265343.12 | 27 | 36 | 16 |
| **2020** | 83/60065.74 | 68/10506.50 | 161/3941.51 | 47/30643.28 | 51/240102.09 | 75/146131.42 | 17 | 48 | 12 |

Sources were found to dominate Green Infrastructure network development, and corridors and codes connected to them to provide material exchange. Due to Hangzhou urban construction and development, overall source areas have declined, the combination degree has become lower, patch fragmentation has grown serious, and biodiversity conservation has deteriorated. First-level sources were found to have a large area and high connectivity, which allowed them to maintain a high level of spatial priority, resulting in a gradually slowing rate of area reduction. For these sources, we recommend strengthening source quality protection, establishing a buffer zone to reduce human activity interference, and expanding their areas. After the of continuous improvement of connections with first-level sources, the advantages of second-level sources gradually appeared. We recommend integrating broken resources with second-level sources and strengthening connections with surrounding sources. Third-level sources were found to have a low spatial priority and small and scattered areas, contributing less to landscape connectivity than other-level sources. Although the total number of third-level sources was always the largest, the state was shown to be unstable and susceptible. We recommend stabilizing the environment around third-level sources, maintaining them for a long time, and gradually improving them.

*3.2. GI Corridors Spatial Prioritization*

Bridges are important channels for core connection, and branches also have connection functions that can be used to extract important data of current corridors. Combining potential corridors revealed that the corridor of $G_{ab} < 10$ was used as a third-level corridor, the corridor of $10 \leq G_{ab} < 100$ was used as a second-level corridor, and the corridor of $G_{ab} \geq 100$ was used as a first-level corridor (Figure 6, Tables 3 and 4). In the past 20 years, corridor construction in the study area mainly comprised second-level and third-level corridors, as well as fewer first-level corridors. The number of first-level corridors increased from 29 to 47, and their length increased from 17,270.50 to 30,643.28 m. The number of second-level corridors increased from 31 to 51, and their length increased from 88,262.73 to 240,102.09 m. Third-level corridors' dominant position decreased from 115 to 75, and their length decreased from 330,152.76 to 146,131.42 m. In some areas, dominant sources were found to be broken, and first-level and second-level corridors' effective connections increased, such as the West Lake Scenic Area (B) and the Shiniu Mountain–Qinghua Mountain Scenic Area (D). Third-level corridors were found to connect to fragmented sources, providing material energy exchange channels for scattered patches in the central and eastern regions, such as the Qiantang River Wetland (C). The spatial priority of some third-level corridors, such as the Banshan National Forest Park (A) and the Jingshan Scenic Area (E), were found to have increased.

Corridors are continuous in form, effectively connect sources at all levels, and generally stable in structure. First-level corridors were found to be distributed around first-level sources, second-level corridor dominance was found to have continuously improved, and third-level corridor advantages as main connecting corridors in the study area were found to have gradually weakened. With first-level source fragmentation, the number and length of first-level and second-level corridors continued to increase, alleviating the crushing rate of first-level sources. Second-level corridors were found to have gradually dominated corridor structure development in the study area due to their higher spatial priority and growing volume. Due the changes in high-spatial-priority corridors and sources, third-level

corridors' spatial priority was found to be continuously improved, and their dominant position gradually declined. The primary protection of first-level corridors, surrounding sources, and ecological land should be used to crush sources and advantageous Green Infrastructure elements in series. Second-level corridor construction should be strengthened to improve biological communication connections, and third-level corridors should be optimized to build effective bridges for material exchange.

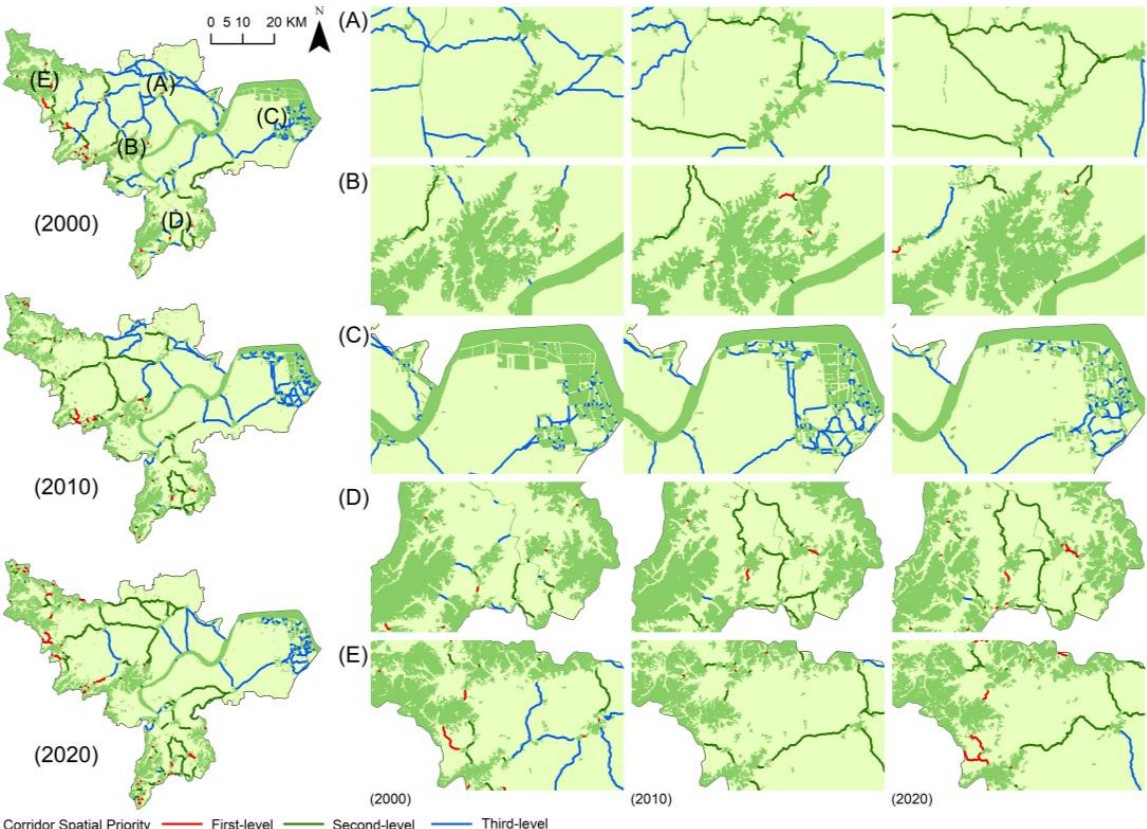

**Figure 6.** GI corridors' spatial priority distribution at all levels in 2000, 2010, and 2020: (**A**) Banshan National Forest Park, (**B**) West Lake Scenic Area, (**C**) Qiantang River Wetland, (**D**) Shiniu Mountain–Qinghua Mountain Scenic Area, and (**E**) Jingshan Scenic Area.

### 3.3. GI Codes' Spatial Prioritization

Codes with $Di < 0.2$ were used as third-level codes, codes with $0.2 \leq Di < 0.5$ were used as second-level codes, and codes with $Di \geq 0.5$ were used as first-level codes (Figure 7, Tables 3 and 4). Codes significantly changed, and distributions tended to be average. First-level codes were found to be mainly distributed in the suburbs, second-level codes were found to engage in material exchange functions at the edge of the city, and third-level codes were found to gradually develop in the middle of the city. The number of first-level codes overall decreased. Due to the Qiantang River Wetland (C) sources' gradual fragmentation, connection corridors were more concentrated, resulting in a dense code concentration in the east. The number of second-level codes increased as a whole and gathered at the edge of the city in areas such as the Shiniu Mountain–Qinghua Mountain Scenic Area (D) and the Jingshan Scenic Area (E). The number of third-level codes overall increased due to ecological construction in the central region, stable corridor structures, and declines in codes' spatial priority, e.g., the West Lake Scenic Area (B).

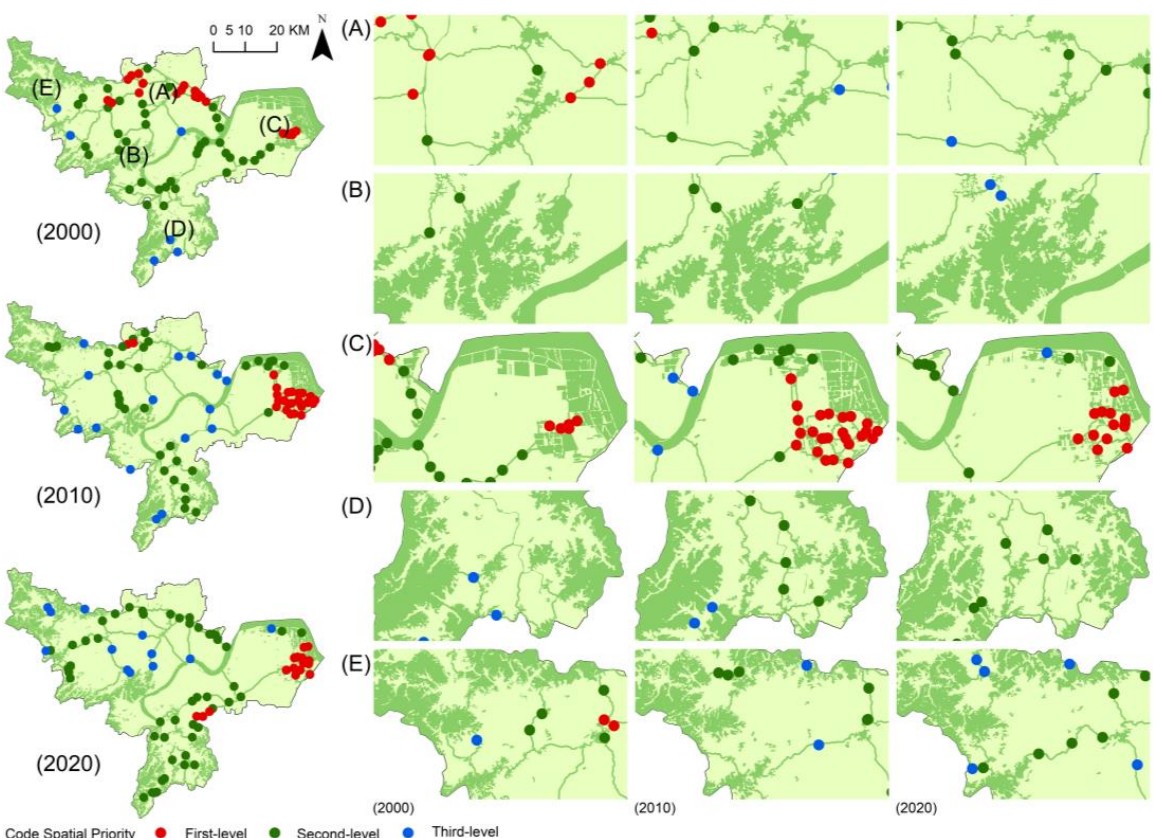

**Figure 7.** GI codes' spatial priority distribution at all levels in 2000, 2010, and 2020: (**A**) Banshan National Forest Park, (**B**) West Lake Scenic Area, (**C**) Qiantang River Wetland, (**D**) Shiniu Mountain–Qinghua Mountain Scenic Area, and (**E**) Jingshan Scenic Area.

Codes' spatial distribution in the study area tended to be average, and their change were significantly compared to corridor change. The spatial prioritization of codes could improve regional landscape pattern connectivity and effectively promote Green Infrastructure network stability. First-level and second-level codes were identified as strategic codes located at main corridor junctions. First-level codes were distributed in clusters, and their spatial changes were obvious. We recommend building first-level codes, increasing their area, and providing stagnant points for biological communication. Second-level codes were found to always occupy dominant positions and have gradually strengthened. Priority should be given to construction on the basis of conservation, which will help improve Green Infrastructure network connectivity and maintain ecological substance circulation. With the close connection of the central corridors, the distribution of second-level codes was found to shift from global to suburb distribution, and third-level codes were found to gradually move closer to the center and weaken in the periphery. Overall, the number of third-level codes was found to increase, thus improving regional connectivity, and future construction needs to be strengthened.

## 3.4. GI Network Spatio-Temporal Evolution

A Green Infrastructure network is organically composed of sources, corridors, and codes. Based on the spatial prioritization analysis, the Green Infrastructure network spatial priority distribution at each level in 2000, 2010 and 2020 was constructed (Figure 8). We found that the Green Infrastructure network spatial structure tended to be stable, sources dominated Green Infrastructure network development, and corridors and codes changed with changing sources. Source patches were seriously fragmented, first-level sources were always in core positions, and the number of second-level sources rapidly increased. Due to first-level source area reductions and broken patches, the number

and length of first-level corridors significantly increased, effectively slowing down the fragmentation rate of sources. Due to weakened first-level sources and optimized third-level sources, the number of second-level corridors and codes was found to have significantly increased. First-level and second-level corridors were found to have gradually grown dense. Second-level corridors were found to have gradually replaced third-level corridors and guided Green Infrastructure network structure development. Due to the low spatial priority of third-level sources, they were found to be unstable and susceptible to urban development, resulting in the disappearance of corresponding connection corridors and landscape connectivity reductions.

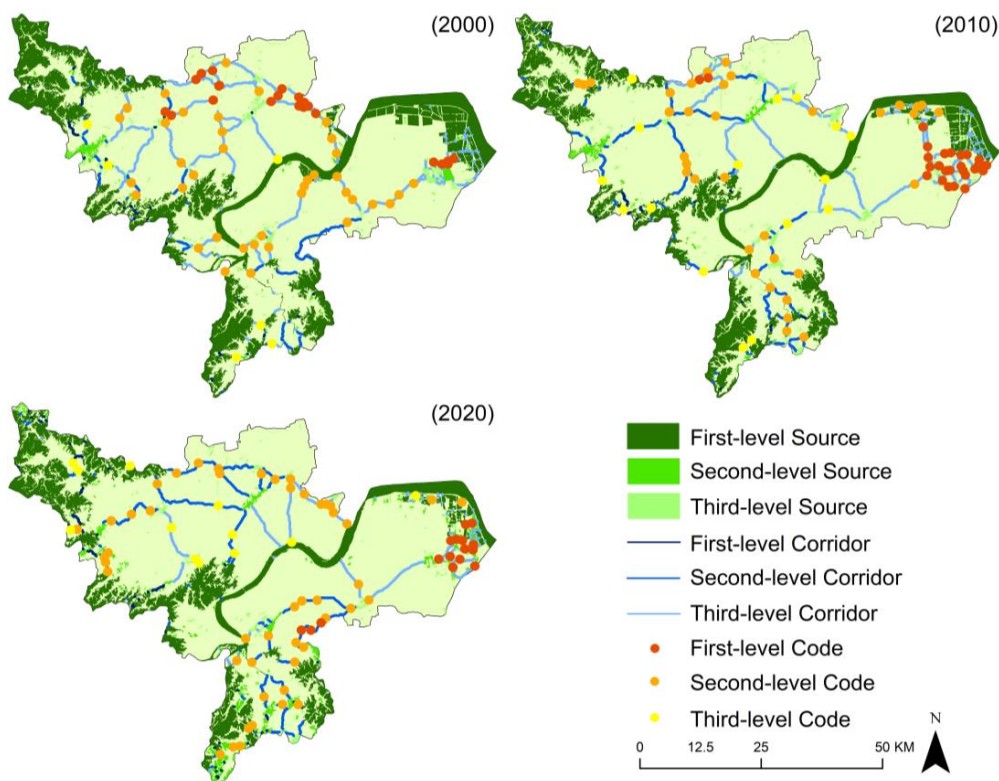

**Figure 8.** GI networks' spatial priority distribution at all levels in 2000, 2010, and 2020.

Due the dual impact of urban development and protection measures, Green Infrastructure network construction in urban areas was found to be effective and Green Infrastructure network structures in suburbs was found to be damaged. In the past two decades, the Hangzhou government has paid more attention to and intensified Green Infrastructure construction in urban areas, which has resulted in gradual increases in Green Infrastructure network spatial priority. Although the area of the network center was found to be small, the surrounding corridors were shown to be dense with evenly distributed morphological continuity and the codes were shown to have been effectively supplemented, resulting in high-quality ecological exchange and strong ecological benefit supplies. Suburban Green Infrastructure networks were found to have been seriously affected by urbanization, and the ecological land adjacent to the urban areas was found to be broken. Green Infrastructure network structure were weakened, and the ecology declined. Due to tourism development and urban construction, sources' spatial priority decreased, the number of corridors gradually decreased, and codes' spatial priority gradually increased. While maintaining the advantages of the Green Infrastructure network in urban areas, we should strengthen suburban Green Infrastructure network construction, and we should effectively balance development and protection in order to achieve Hangzhou's goal of building a suburban natural ecological circle and an urban green living circle.

## 4. Discussion and Conclusions

### 4.1. A New Perspective and Framework of Green Infrastructure Network Construction

Rapid urbanization brought many challenges to human development, brought great pressure to ecological space, and caused a series of ecological problems. As a result, the demand and importance of urban Green Infrastructure construction is rapidly increasing. Optimizing urban green space through Green Infrastructure construction and promoting the sustainable development of cities have become hot research spots. Instead of using the traditional Green Infrastructure network construction method, we quantified Green Infrastructure network changes by identifying and dividing the spatial priority of Green Infrastructure elements. The spatio-temporal characteristic analysis of the spatial priority of Green Infrastructure networks can be used to regulate Green Infrastructure network construction timing and change disorderly construction patterns. Here, we used MSPA and MCR to accurately identify Green Infrastructure elements and to systematize complex landscape patterns, showing that landscape connectivity analysis can be used to effectively spatial prioritize Green Infrastructure elements and construct Green Infrastructure networks. We formed a new research framework of "source determination–resistance surface setting–corridor construction–code identification–spatial prioritization–GI network construction", thus providing a new perspective and framework for the regulation of urban ecological space construction timing.

### 4.2. Reveal the Spatial Distribution of Key GI Elements and GI Network variations

Our findings reveal the spatial distribution of key Green Infrastructure elements that have important implications for Green Infrastructure network stability and connectivity. Hangzhou is rich in ecological resources; they are dense in peripheral areas and sparsely distributed in the central region. Sources were found to dominate Green Infrastructure network development and change, and corridors and codes were shown to connect sources through material exchanges. Hangzhou source patches were found to be seriously fragmented, first-level sources were always found in core positions, and the number of second-level sources was found to have been rapidly increasing. Corridors varied around sources, gradually increasing as plaques broke down. The total number of third-level sources and corridors was the largest, and their spatial priority was low and vulnerable to encroachment. The results show that the spatial priority of Hangzhou Green Infrastructure network has gradually increased and its structure is stable. However, there are still problems such as weak protection that need to be improved. We recommend optimizing Green Infrastructure networks by strengthening Green Infrastructure elements' construction. Additionally, there are differences in the development of urban and suburban Green Infrastructure networks, so we recommend strengthening suburban Green Infrastructure network structure construction, maintaining urban Green Infrastructure network advantages, and effectively balancing development and protection.

### 4.3. GI Network Construction, Optimization and Differentiated Governance

The quantitative analysis of the spatio-temporal characteristics and spatial priority of Green Infrastructure network is helpful for guiding construction decision by category and level, as well as guiding Green Infrastructure network ecological protection and planning prioritization. The construction of Green Infrastructure networks in chronological and hierarchical levels according to spatial priorities can provide high-quality ecological service functions. This process can be used to gradually improve the urban ecological environment and to improve the ability of urban sustainable development. Green Infrastructure networks should be gradually built by means of protection, system repair, and rational utilization to ensure the priority construction of key areas and differentiated management. In addition, the connotations and characteristics of different Green Infrastructure elements should be considered in this governance approach, under which source quality protection should be strengthened, broken resources should be integrated through ecological restoration and connection, quantity and efficiency should be improved, corridor structure

should be optimized, corridor width should be broadened, Green Infrastructure network structure stability should be improved, biological exchanging connectivity and construction efficiency should be improved, code construction should be improved, transfer station functions should be serious considered, area should be increased, biological communication stagnation points for longer corridors should be provided, and Green Infrastructure networks' overall connectivity should be increased. Against the background of rapid urbanization and land resource scarcity, these strategies can be used to effectively restore and build cities' ecological spaces.

### 4.4. Limitations and Uncertainties

Although this study provides new insights into the construction and protection of Green Infrastructure networks due to its targeted analysis of the spatio-temporal characteristics and spatial prioritization of Green Infrastructure networks, it also had limitations and uncertainties. First, in this study, we used 30 x 30 m land-use data, not high-score series data. Due to the used data resolution, some small-scale ecological land may not have been identified as GI data. Second, the specifics of our Green Infrastructure research process, especially regarding the determination of indicators and thresholds, impacted our results. The parameter settings of MSPA and connectivity, as well as the influence of social factors on resistance construction, can cause theoretical Green Infrastructure research to differ from reality. Finally, we ignored the impact of humanistic functions such as leisure, recreation, and cultural inheritance on the construction of urban Green Infrastructure. Follow-up research needs to be further deepened and improved.

**Author Contributions:** Conceptualization, X.C. and L.X.; Funding acquisition, X.C., L.X., Q.M. and Y.S.; Investigation, R.Z.; Methodology, X.C. and L.X.; Supervision, L.X., R.Z., Q.M., Y.S. and Z.L.; Visualization, X.C.; Writing—original draft, X.C., L.X., Q.M., Y.S. and Z.L.; Writing—review and editing, X.C., L.X. and Y.S. All authors have read and agreed to the published version of the manuscript.

**Funding:** This research was funded by the Key Program of Zhejiang Province Philosophy and Social Science Planning Interdisciplinary (No. 22JCXK06Z), Zhejiang Provincial Natural Science Foundation (No. LQ22E080007), General Program of National Natural Science Foundation of China (No. 41871216), Zhejiang Province Natural Science Foundation of China (No. LQ20D010002), and Zhejiang Provincial College Student Science and Technology Innovation Plan and Planted Talent Plan Funding Project (No. 2021R412046).

**Institutional Review Board Statement:** Not applicable.

**Informed Consent Statement:** Not applicable.

**Data Availability Statement:** All data used in this study are issued by US Geological Survey. This data can be found here: (http://glovis.Usgs.gov/, accessed on 10 November 2021).

**Acknowledgments:** The authors gratefully acknowledge the support of the funding.

**Conflicts of Interest:** The authors declare no conflict of interest.

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
