# Peer review of "Changes and Characteristics of Green Infrastructure Network Based on Spatio-Temporal Priority"

_land, doi:10.3390/land11060901_

Round 1
Reviewer 1 Report
Improve the conclusion paragraph
Author Response
Point 1: Improve the conclusion paragraph
Response 1: Thanks for reviewer’s suggestion. We have adjusted the structure of the article, merging the contents of discussion and conclusion. On the basis of the previous, the content of conclusions obtained is supplemented. We divide the discussion and conclusion section into four sections.The details are as follows:
1) A new perspective and framework of Green Infrastructure network construction
Rapid urbanization brought many challenges to human development, brought great pressure to ecological space, and caused a series of ecological problems. As a result, the demand and importance of urban Green Infrastructure construction is increasing rapidly. How to optimize urban green space through GI construction and promote the sustainable development of cities has become a hot spot. Different from traditional Green Infrastructure network construction method, Green Infrastructure network changes can be shrewdly quantified through identifying and dividing spatial priority of Green Infrastructure elements to building Green Infrastructure network. The spatio-temporal characteristics analyzation of Green Infrastructure network spatial priority can possible to shrewdly regulate Green Infrastructure network construction timing and change disorderly construction pattern. Here, we use MSPA and MCR to accurately identify Green Infrastructure elements and systematize complex landscape patterns. Use landscape connectivity analysis to effectively spatial prioritize of Green Infrastructure elements and construct Green Infrastructure network. We form a new research framework of "source determination-resistance surface setting-corridor construction-code identification-spatial prioritization-GI network construction". Provide a new perspective and framework for shrewd regulation of urban ecological space construction timing.
2) Reveal the spatial distribution of key GI elements and GI network variations
Our finding reveals the spatial distribution of key Green Infrastructure elements that have important implications for Green Infrastructure network stability and connectivity. Hangzhou ecological resources is rich, dense in the periphery and sparsely distributed in the central region. Sources dominates Green Infrastructure network development and change, and corridors and codes connect sources through material exchanges. Hangzhou source patches are seriously fragmented, first-level sources are always in core position, and the number of second-level sources is increasing rapidly. Corridors vary around sources, gradually increasing as plaques break down. Total number of third-level sources and corridors is the largest, and their spatial priority is low and vulnerable to encroachment. The result shows that the spatial priority of Hangzhou Green Infrastructure network has gradually increased and its structure is stable. But there are still problems such as weak protection and need to be improved. Optimize Green Infrastructure network by strengthening Green Infrastructure elements construction. There are differences in the development of urban and suburban green infrastructure networks. Strengthen suburban GI network structure construction, maintain urban GI networks advantages, and effectively balance development and protection.
3)GI network construction, optimization and differentiated governance
Quantitative analysis spatio-temporal characteristics of GI network spatial priority is helpful to the proposal of construction by category and level, playing a role in guiding Green Infrastructure network ecological protection and planning prioritization. The construction of GI networks in chronological and hierarchical levels according to spatial priorities can provide high-quality ecological service functions. It can gradually improve the urban ecological environment, and improve the ability of urban sustainable development. Gradually build Green Infrastructure network by means of protection, system repair, and rational utilization, to ensure priority construction of key areas and differentiated management. . In addition, the connotations and characteristics of different GI elements are also taken into account in governance approach. Strengthen sources quality protection, integrate broken resources through ecological restoration and connection, and improve quantity and efficiency. Optimize corridors structure, broaden corridor width, improve Green Infrastructure network structure stability, improve biological exchanging connectivity and construction efficiency. Improve codes construction, give full play to transfer station function, increase area, provide biological communication stagnation points for longer corridors, and increase Green Infrastructure network overall connectivity. On the background of rapid urbanization and land resources scarcity, better effectively restore and build city ecological space.
4)Limitations and Uncertainties
Although this study provides new insights into the construction and protection of Green Infrastructure network, and conducts targeted analysis of the spatio-temporal characteristics of Green Infrastructure network spatial prioritization, it also comes with limitations and uncertainties. First, this study uses the accuracy of 30*30m land use data for research, not high-score series data. Due to data accuracy, the missing problem may be happen where some small-scale ecological land is not identified and GI data. Second, there are differences in the determination of indicators and thresholds in the Green Infrastructure research process, which will finally have a certain impact in Green Infrastructure. The parameter setting of MSPA and connectivity, as well as the influence of social factors on resistance construction, will cause Green Infrastructure to differ from reality. Finally, this study ignores the impact of humanistic functions such as leisure and recreation and cultural inheritance on the construction of urban Green Infrastructure. The follow-up research needs to be further deepened and improved.
In addition, during the process of revising the manuscript, we used the Track Changes feature of the word software to make it easier for the editor-in-chief and reviewers to clearly understand the revisions.
Thanks again to the reviewers for their helpful comments and the editors for their help in this process. Looking forward to your reply.
Best wishes.

Reviewer 2 Report
According to Figure 2, it is perceived that a very harsh land use change –especially for changing Farmlands into Buildings- has been occurred. Although the forests kept rather unchanged, changing Farmlands to Buildable Lands is too abrupt annoying the sustainability aims. What have been the main scientific reasons for such a huge change? From other hand, as years 2000, 2010 and 2020 for study; this involves temporary assessment so it is better to import the term “Temporary analyses” into title.
This article is so related to Sustainability and Sustainable Development; however, it is dealt with these matters very little. I highly advise to define and insert some Sustainability Indicators SI for engaging the spatial priority with sustainability aims. Such Harsh changes as that of Figure 2 assign deviations from sustainability which must be studied again and again.
It is seemed that the article methodology and its results are exclusive for the case study. For applicability of the article it is highly advised to generalize and enlarge the methods for other case studies. It is better to define Indices for sustainability, each GI element extract and Spatial and Temporal Prioritization with scaled outputs. These indices should have defined and clear metric output levels for being capable to be applied for other places. Giving some examples for calculation of equations helps very much. For example how interaction forces are calculated or measured? Or how Corridor Resistances are allocated? After Defining and scaling your indices Please prepare clear Tables or Roadmaps for your methodological Indices. I mean every element -either input or output- in your indices should be scaled and degreed with clear and defined levels by proper names like following:
· High, Average, Low Interaction Forces with cross points
· High, Average, Low Sustainability Indices with Cross points
· High, Average, Low Spatial or Temporal Prioritization Indices with Cross points
· As well for all metric indices in your study
Rearranging the title as something like following can help very much in receiving high impact and achieving increasing citations: “Novel Indices for Changes and Characteristics of Green Infrastructure Network Based on Spatio-Temporal Priority and Sustainability”.
Finally I found the article is well written and can receive acceptance after above notes be performed.
Author Response
Point 1: According to Figure 2, it is perceived that a very harsh land use change – especially for changing Farmlands into Buildings - has been occurred. Although the forests kept rather unchanged, changing Farmlands to Buildable Lands is too abrupt annoying the sustainability aims. What have been the main scientific reasons for such a huge change?
Response 1: Thanks for reviewer’s suggestions. Land-use coverage is undergoing intense change under rapid urbanization. China has been on a fast track of land-use change since the Reform and Opening-up policy in 1978. Relevant research proves that the farmland and the buildable land changed in a large proportion during the study period in China, especially in rapid urban development areas. Since the beginning of the 20th century, the increase of land area alleviated the contradiction of supply and demand between human and land, which provided the guarantee for agricultural production and industrial development in Hangzhou.(https://www.mdpi.com/1660-4601/16/7/1124) Based on your suggestions and our thoughts on this, we have supplemented the line 288 to 290 section of the article. The details are as follows:
From 2000 to 2020, Hangzhou was in a rapid development period, and the urban built-up area was expanding rapidly. Excessive encroachment of buildable land on farmland has led to drastic changes in land use.
Point 2: This article is so related to Sustainability and Sustainable Development; however, it is dealt with these matters very little. I highly advise to define and insert some Sustainability Indicators SI for engaging the spatial priority with sustainability aims. Such Harsh changes as that of Figure 2 assign deviations from sustainability which must be studied again and again.
Response 2: Thanks for reviewer’s suggestions. Sustainability is indeed important, especially for the construction of ecological corridors, the prioritization of spaces and the optimization of Green Infrastructure networks. We did consider some of the impacts of sustainability in our research. In the process of building the resistance model, we mainly considered the natural factors and urban construction conditions that affect the sustainable development of the city. We screened for resistance factors of sustainable significance in our study, MSPA Landscape Type, Land Use Type, Elevation, Slope, Relief Degree of Land Surface.
We also discussed sustainability in the discussion and conclusion section. On the one hand, Green Infrastructure network building has sustainable implications. Rapid urbanization brought many challenges to human development, brought great pressure to ecological space, and caused a series of ecological problems. As a result, the demand and importance of urban Green Infrastructure construction is increasing rapidly. How to optimize urban green space through GI construction and promote the sustainable development of cities has become a hot spot. The spatio-temporal characteristics analyzation of Green Infrastructure network spatial priority can possible to shrewdly regulate Green Infrastructure network construction timing and change disorderly construction pattern. On the other hand, the construction and optimization of Green Infrastructure networks follow the principles of sustainable development. Quantitative analysis spatio-temporal characteristics of GI network spatial priority is helpful to the proposal of construction by category and level, playing a role in guiding Green Infrastructure network ecological protection and planning prioritization. The construction of GI networks in chronological and hierarchical levels according to spatial priorities can provide high-quality ecological service functions. It can gradually improve the urban ecological environment, and improve the ability of urban sustainable development. Gradually build Green Infrastructure network by means of protection, system repair, and rational utilization, to ensure priority construction of key areas and differentiated management. On the background of rapid urbanization and land resources scarcity, better effectively restore and build city ecological space.
Considering the importance of sustainable development, the next step we will be to combine spatial priorities with sustainability goals from the perspective of sustainable development in subsequent studies. We will continue to deepen the deeper issues such as the impact mechanism of sustainable development.
Point 3: It is seemed that the article methodology and its results are exclusive for the case study. For applicability of the article it is highly advised to generalize and enlarge the methods for other case studies. It is better to define Indices for sustainability, each GI element extract and Spatial and Temporal Prioritization with scaled outputs. These indices should have defined and clear metric output levels for being capable to be applied for other places. Giving some examples for calculation of equations helps very much. For example how interaction forces are calculated or measured? Or how Corridor Resistances are allocated? After Defining and scaling your indices Please prepare clear Tables or Roadmaps for your methodological Indices. I mean every element -either input or output- in your indices should be scaled and degreed with clear and defined levels by proper names like following:
ï‚Ÿ High, Average, Low Interaction Forces with cross points
ï‚Ÿ High, Average, Low Sustainability Indices with Cross points
ï‚Ÿ High, Average, Low Spatial or Temporal Prioritization Indices with Cross points
ï‚Ÿ As well for all metric indices in your study
Response 3: Thanks for reviewer’s suggestions. 1) This study mainly studies the typical areas of rapid urbanization, adopts the spatial prioritization of green infrastructure network to optimize Green Infrastructure networks, and proposes to carry out phased construction of green infrastructure networks of different levels. The purpose of the study is to seek solutions for smart and coordinated urban growth and ecological protection in developing country urbanized areas. In China, this type of city mainly includes Beijing, Shanghai, Hangzhou and other cities. Since the 20th century, with people influx and continuous urbanization acceleration, Hangzhou has developed rapidly. According to data, Hangzhou population urbanization rate rose to 83.29% in 2020, the total GDP in 2020 has an increase of 3.9% over the previous year, and the built-up area by 2020 was 567.32 km2. Under the background of limited land resources, as Hangzhou important part, ecological risks brought about by rapid urbanization are particularly prominent in the main city zone. Therefore, taking Hangzhou as the research object is very typical. The study has certain reference significance for the spatio-temporal characteristics analysis of green infrastructure networks in other regions, and can provide an important reference and basis for green infrastructure networks construction.We have made detailed additions to this at line 228 to 232 and line 255-264.
2) We have supplemented the content based on your comments to improve the applicability of the article. We have added Table 2. Resistance values and resistance coefficients of each factor from line 344, which clarifies the composition of the resistance factors and the setting of weights for each resistance layer. At the same time, we have added the Table 3. Division and definition of different GI elements spatial prioritization at each level from line 421, which clarifies the classification criteria and definition for the spatial priority of different Green Infrastructure elements at different levels, and is convenient for reference application in other places. Based on your recommendations and the meaning of Green Infrastructure elements of different spatial priority, we define the first-level spatial priority Green Infrastructure elements as High Spatial or Temporal Prioritization sIndicesin Landscape Connectivity, the Second-level spatial priority Green Infrastructure elements as Average Spatial or Temporal Prioritization Indicesin Landscape Connectivity and the Second-level spatial priority Green Infrastructure elements as Low Spatial or Temporal Prioritization Indicesin Landscape Connectivity.
Point 4: From other hand, as years 2000, 2010 and 2020 for study; this involves temporary assessment so it is better to import the term “Temporary analyses” into title. Rearranging the title as something like following can help very much in receiving high impact and achieving increasing citations: “Novel Indices for Changes and Characteristics of Green Infrastructure Network Based on Spatio-Temporal Priority and Sustainability”.
Response 4: Thanks for reviewer’s suggestions. The study in this paper is indeed involving temporary assessment of green infrastructure networks for three years. The current title really does not reflect temporary assessment. Your suggestions for title remodeling adjustments are very good and have a qualitative improvement. Some Indices were indeed used in the study, considering that the relevant indicators are widely used in the study of green space. Since we are less innovative and not deep enough in the indicator, it may not be very suitable to add " Novel Indices " to the title.
Sustainability does require ongoing discussion, which is after the study, so not much research has been done in this article. Our current research target is the characterization and optimization of Green Infrastructure network spatial prioritization. The in-depth discussion of sustainability may be another research question, which inspired me to conduct a holistic study of Green Infrastructure networks from a sustainability perspective in subsequent studies. Combined the revision of the article with your suggestion to modify the title to Changes and Characteristics of Green Infrastructure Network Based on Spatio-Temporal Priority.
In addition, during the process of revising the manuscript, we used the Track Changes feature of the word software to make it easier for the editor-in-chief and reviewers to clearly understand the revisions.
Thanks again to the reviewers for their helpful comments and the editors for their help in this process. Looking forward to your reply.
Best wishes.

Reviewer 3 Report
Reviewer Comments:
General comments:
It is my pleasure to review the paper entitled “Opportunities for Using Analytical Hierarchy Process in Green Architecture / Green Building Optimization”. This is interesting and selects the main city zone in Hangzhou as the study area, extracting 2000, 2010 and 2020 land use data. Use Morphological Spatial Pattern Analysis to identify Green Infrastructure landscape types and further extract Green Infrastructure elements. Identify Green Infrastructure network spatial priority through landscape connectivity evaluation according to ecological importance and development vulnerability. The paper is within the scope of Land. However, it needs some major changes before publication.
I present the following comments that can help to improve the paper:
Please check the title and revise it.
Detailed comments:
Abstract
The authors should write the main problem and then start how this study is useful to tackle those issues.
Introduction
- Line 34-37 Reference?
- Line 46-47. Kindly extend the Introduction part by providing recent references from recent articles in the introduction section. i.e. https://doi.org/10.3390/w10040435; https://doi.org/10.3390/su14074247; https://doi.org/10.3390/su10030584; https://doi.org/10.1016/j.scs.2022.103798.
Other comments:
I have a few concerns.
1.What was your contribution specifically in this area? what is the accuracy of MSPA?. The following are more detailed comments on each.
2. What are your research questions and hypotheses? What are the important findings expected by the readers? What are research gaps in the past, and what are your contributions to improve them?
3. As the study include 2020 data? How about the impact of COVID 19 etc.?
4. As it is mentioned that Mapping Spatiotemporal Distributions of Green Infrastructure at Building Scale, so I wonder how the author considers the different types of buildings (orientation, design, height, and other variables for comprehensive assessment) for the result analysis?
5. How the accuracy was assessed? please elaborate.
Best,
Author Response
Point 1: Please check the title and revise it.
Response 1: Thanks for reviewer’s suggestion. The study in this paper involves a three-year temporary assessment of green infrastructure networks. The current title does not actually reflect the temporary assessment. Combined the revision of the article with your suggestion to modify the title to Changes and Characteristics of Green Infrastructure Network Based on Spatio-Temporal Priority.
Point 2: The authors should write the main problem and then start how this study is useful to tackle those issues in the abstract.
Response 2: Thanks for reviewer’s suggestions. With advancement urbanization, natural lands are constantly encroached upon by artificial im-pervious surface, accompanied, and ecosystem is seriously damaged. Calls for green infrastructure to address urban environmental issues and resource reallocation are growing. The main problem is that how to optimize green infrastructure network construction in under rapid urbanization. The study identified Green Infrastructure network elements spatial priority through landscape connectivity evaluation according to ecological importance and development vulnerability. After the construction of Green Infrastructure network, analyze the spatio-temporal characteristics of Green Infrastructure network spatial priority. Through the spatial prioritization of Green Infrastructure networks, the gradual construction and optimizion of Green Infrastructure networks will help to improve urban green spaces in stages. The Abstract section was modified according to your suggestions. The details are as follows:
With advancement urbanization, natural lands are constantly encroached upon by artificial im-pervious surface, accompanied, and ecosystem is seriously damaged. Calls for green infrastructure to address urban environmental issues and resource reallocation are growing. How to optimize Green Infrastructure networks is becoming increasingly important under rapid urbanization. The study takes the main city zone in Hangzhou as study area, extracting 2000, 2010 and 2020 land use data. Use Morphological Spatial Pattern Analysis to identify Green Infrastructure landscape types and further extract Green Infrastructure elements. Identify Green Infrastructure network elements spatial priority through landscape connectivity evaluation according to ecological importance and development vulnerability. After the construction of Green Infrastructure network, analyze the spatio-temporal characteristics of Green Infrastructure network spatial priority. Through the spatial prioritization of Green Infrastructure networks, the gradual construction and optimizion of Green Infrastructure networks will help to improve urban green spaces in stages. Smartly coordinate urban growth and ecological protection based on Green Infrastructure spatial prioritization, which may help improve urban living environment and enhance sustainable urban development capabilities. The conclusion is that sources dominate corridors and codes changing. Sources is fragmented, the integration degree become lower, and first-level sources advantage is weakened. The corridors morphology develops continuously and corridor structure stabilizes. Second-level corridors gradually replace third-level corridors to guide Green Infrastructure network structures development. Codes distribute scatter and tend to average, closely following corridors change.
Point 3: Line 34-37 Reference? Line 46-47. Kindly extend the Introduction part by providing recent references from recent articles in the introduction section. i.e.
Response 3: Thanks for reviewer’s suggestions. We carefully read several of the articles you provided and thought deeply about them. We found that these articles do have great reference value for the research of this article, and can support and improve the content of the article. So we went into the article with references. You can find our new reference from Line 39-42 and Line 59-60. Based on your comments, we have also added some support for the relevant literature for other parts of the article. We have added corresponding references to Line 313-317.
[8]Shafique M, Kim R. Recent progress in low-impact development in South Korea: Water-management policies, challenges and opportunities[J]. Water, 2018, 10(4): 435.
[9]Newman G, Sansom G T, Yu S, et al. A Framework for Evaluating the Effects of Green Infrastructure in Mitigating Pollutant Transferal and Flood Events in Sunnyside, Houston, TX[J]. Sustainability, 2022, 14(7): 4247.
[10]Li X, Stringer L C, Dallimer M. The role of blue green infrastructure in the urban thesrmal environment across seasons and local climate zones in East Africa[J]. Sustainable Cities and Society, 2022, 80: 103798.
[15]Wang J, Banzhaf E. Towards a better understanding of Green Infrastructure: A critical review[J]. Ecological Indicators, 2018, 85: 758-772.
[60]Shi X, Qin M. Research on the optimization of regional green infrastructure network[J]. Sustainability, 2018, 10(12): 4649.
[61]Wickham J D, Riitters K H, Wade T G, et al. A national assessment of green infrastructure and change for the conterminous United States using morphological image processing[J]. Landscape and Urban Planning, 2010, 94(3-4): 186-195.
Point 4: What is the accuracy of MSPA? How the accuracy was assessed? please elaborate.
Response 4: Thanks for reviewer’s suggestions. The type of landscape acquired based on the MSPA had a clear scale effect and edge effect. On the one hand, the increase in scale will lead to the loss of spatial information, and as the scale increases, the MSPA landscape types are unstable and in a state of dynamic change. The greater the spatial accuracy of the input data, the more detailed information about the landscape type can be obtained. Due to data amount, this study uses remote sensing image data of Landsat TM/OLI and the input data of pixel size 30 m, which is facilitate data processing at higher resolutions. The input data of MSPA is a binary plot translated after the classification of land use. Therefore, we also consider the accuracy of land classification. The kappa coefficients (K) of three years land use classification results are greater than 0.8. On the other hand, the change of edge width will not lead to the loss of spatial information and the change of the landscape types, but will only lead to the internal transformation of the landscape types. The smaller the edge width, the more detailed information about the type of landscape can be obtained. Comprehensive previous research and multiple sets parameters experimental comparison, use eight-neighborhood rule and set edge width to one. The final use of MSPA is the result of higher accuracy and detailed spatial information. We have added some additions to this section from line 281 to 283, line 287 to 288 and line 312 to 317. At the same time, we added Figure 4. MSPA landscape types of the study area in 2000, 2010, and 2020 from line 325.
Point 5: What are your research questions and hypotheses? What are the important findings expected by the readers? What are research gaps in the past, and what are your contributions to improve them?What was your contribution specifically in this area?
Response 5: Thanks for reviewer’s suggestions. Based on your questions, we have perfected and added some supplement additions for the abstract, introduction, discussion and conclusion part.With advancement urbanization, natural lands are constantly encroached upon by artificial im-pervious surface, accompanied, and ecosystem is seriously damaged. Calls for green infrastructure to address urban environmental issues and resource reallocation are growing. Smartly coordinate urban growth and ecological protection based on Green Infrastructure spatial prioritization, which may help improve urban living environment and enhance sustainable urban development capabilities. How to optimize Green Infrastructure networks is becoming increasingly important under rapid urbanization.
Our research question is the spatial-temporal variation characteristics of Green Infrastructure, to gradually improve urban green space through the step-by-step construction of green infrastructure network. This study can also bring some important findings to the reader. We reveal the spatial distribution of key GI elements and GI network variations. Our finding reveals the spatial distribution of key Green Infrastructure elements that have important implications for Green Infrastructure network stability and connectivity.At the same time, we propose GI network construction, optimization and differentiated governance. Quantitative analysis spatio-temporal characteristics of Green Infrastructure network spatial priority is helpful to the proposal of construction by category and level, playing a role in guiding Green Infrastructure network ecological protection and planning prioritization. Gradually build Green Infrastructure network by means of protection, system repair, and rational utilization, to ensure priority construction of key areas and differentiated management. Therefore, the study has certain reference significance for the spatio-temporal characteristics analysis of green infrastructure networks in other regions, and can provide an important reference and basis for green infrastructure networks construction.
The existing research is mainly expected that the optimization path of Green Infrastructure network construction will be proposed from the change of plot overall index. However, differentiating management strategies for different Green Infrastructure spatial priorities can ensure that limited land resources and public funds are directed to where they are truly needed. Through the spatial prioritization of green infrastructure network, this study can play a role in shrewdly regulating urban construction and optimizing green infrastructure. At the same time, our research complements the research structure of green infrastructure network construction. Here, we use MSPA and MCR to accurately identify Green Infrastructure elements and systematize complex landscape patterns. Use landscape connectivity analysis to effectively spatial prioritize of Green Infrastructure elements and construct Green Infrastructure network. We form a new research framework of "source determination-resistance surface setting-corridor construction-code identification-spatial prioritization-GI network construction". Provide a new perspective and framework for shrewd regulation of urban ecological space construction timing.
Point 6: As the study include 2020 data? How about the impact of COVID 19 etc.?
Response 6: Thanks for reviewer’s suggestions. COVID 19 does have a certain impact on urbanization, especially in terms of population urbanization. However, since this paper mainly focuses on the spatio-temporal characteristics of green infrastructure networks, the main influencing factor is the process of land urbanization. In China, COVID 19 has not stopped the process of land urbanization, or the impact is less obvious. The impact on small-scale research targets may be large, and it is not obvious at the regional level and can be ignored. Your good suggestions also inspired me to consider the impact of the COVID 19 in the follow-up small-scale green infrastructure research, and to respond to public emergencies such as the epidemic in the construction of green infrastructure network.
Point 7: As it is mentioned that Mapping Spatiotemporal Distributions of Green Infrastructure at Building Scale, so I wonder how the author considers the different types of buildings (orientation, design, height, and other variables for comprehensive assessment) for the result analysis?
Response 7: Thanks for reviewer’s suggestions.This study mainly conducts the spatio-temporal characteristics of green infrastructure network from the regional scale. Research from the architectural scale of green infrastructure is a micro study, and the different types of buildings (orientation, design, height, and other variables for comprehensive assessment) does need to be considered in the follow-up. The impact of buildings on green infrastructure may be mainly present in the roof garden and the smaller green space around it. On the one hand, roof gardens and gray buildings are opposites, and roof gardens can eliminate the damage of some buildings to green spaces, but this is often overlooked in large-scale studies. On the other hand, the smaller green space around the building, in the study, due to the problem of data accuracy, there is a problem of neglect, therefore, resulting in the incompleteness of the study. Your advice has taught me a huge lesson in taking into account the impact of buildings in small-scale research.
In addition, during the process of revising the manuscript, we used the Track Changes feature of the word software to make it easier for the editor-in-chief and reviewers to clearly understand the revisions.
Thanks again to the reviewers for their helpful comments and the editors for their help in this process. Looking forward to your reply.
Best wishes.

Round 2
Reviewer 3 Report
The paper can be accepted for publication.
Best,